# Radiomics and Machine Learning in Brain Tumors and Their Habitat: A Systematic Review

**DOI:** 10.3390/cancers15153845

**Published:** 2023-07-28

**Authors:** Mehnaz Tabassum, Abdulla Al Suman, Eric Suero Molina, Elizabeth Pan, Antonio Di Ieva, Sidong Liu

**Affiliations:** 1Centre for Health Informatics, Australian Institute of Health Innovation, Macquarie University, Sydney, NSW 2109, Australia; sidong.liu@mq.edu.au; 2Computational NeuroSurgery (CNS) Lab, Macquarie Medical School, Macquarie University, Sydney, NSW 2109, Australia; abdulla.suman@mq.edu.au (A.A.S.); eric.suero-molina@mq.edu.au (E.S.M.); elizabeth.pan@students.mq.edu.au (E.P.); 3Department of Neurosurgery, University Hospital of Münster, 48149 Münster, Germany; 4Faculty of Medicine and Health Sciences, Macquarie University, Sydney, NSW 2109, Australia

**Keywords:** radiomics, brain tumor, peritumoral region, tumor habitat, neuro-oncology, machine learning

## Abstract

**Simple Summary:**

Radiomics involves the extraction of quantitative features from medical images, which can provide more detailed and objective information about the features of a tumor compared to visual inspection alone. By analyzing the extensive range of features obtained through radiomics, machine-learning techniques can enhance tumor diagnosis, assess treatment response, and predict patient prognosis. This review highlights the mutual impact between the tumor and its microenvironment (habitat), in which tumor cells can modify the microenvironment to promote their growth and survival. At the same time, the habitat can also influence the behavior of tumor cells. Encouragingly, radiomics and machine learning have shown promising potential in diagnosing brain tumors and predicting prognosis. However, several limitations still need to be improved for their practical application in clinical settings. Further research is required to optimize radiomic feature extraction, standardize imaging protocols, validate models on larger datasets, and integrate diverse data to facilitate a more comprehensive analysis.

**Abstract:**

Radiomics is a rapidly evolving field that involves extracting and analysing quantitative features from medical images, such as computed tomography or magnetic resonance images. Radiomics has shown promise in brain tumor diagnosis and patient-prognosis prediction by providing more detailed and objective information about tumors’ features than can be obtained from the visual inspection of the images alone. Radiomics data can be analyzed to determine their correlation with a tumor’s genetic status and grade, as well as in the assessment of its recurrence vs. therapeutic response, among other features. In consideration of the multi-parametric and high-dimensional space of features extracted by radiomics, machine learning can further improve tumor diagnosis, treatment response, and patients’ prognoses. There is a growing recognition that tumors and their microenvironments (habitats) mutually influence each other—tumor cells can alter the microenvironment to increase their growth and survival. At the same time, habitats can also influence the behavior of tumor cells. In this systematic review, we investigate the current limitations and future developments in radiomics and machine learning in analysing brain tumors and their habitats.

## 1. Introduction

Brain cancer is responsible for a significant number of deaths, ranking among the ten most common causes of cancer-related death [1]. These tumors can be primary or metastatic in nature. Approximately 80% of primary malignant brain tumors are classified as gliomas, encompassing several subtypes, including astrocytoma, oligodendroglioma, ependymoma, and the most malignant type, glioblastoma (GBM) [2,3]. The distinctive traits of specific subtypes of glioma, such as invasive and proliferative cell behaviour, angiogenesis, apoptosis, and significant heterogeneity, collectively contribute to heightened morbidity and mortality rates. Of these subtypes, GBM is the most aggressive and diverse type, displaying high heterogeneity in cellular and molecular characteristics, and it leads to low levels of short-term survival [4]. The average 5-year survival rate for GBM is between 5.6% and 7%, with a median survival period of approximately 12 to 15 months [5]. Although patients receive intensive treatment through surgical procedures, radiotherapy, and chemotherapy, the overall survival rate remains disheartening [6]. The early diagnosis and accurate classification of brain tumors can facilitate prompt treatment and help to prevent further tumor growth and spread, improving the effectiveness of therapy and survival rates for patients.

Neuro-oncology relies on magnetic resonance imaging (MRI) as the preferred method for diagnosing, evaluating treatment response, and predicting prognosis. This non-invasive technique provides extensive information about tumors and peritumoral regions [7]. There is an increasing interest in the use of radiomics to extract reproducible features from images, including complex patterns not visible to the human eye [8,9]. By examining the statistical inter-relationships between voxels, these features may reflect the underlying dynamics of smaller-scale biological phenomena or disease pathophysiology [10]. Therefore, they are widely used to describe brain tumors’ radiologic phenotypes [11]. 

Radiomics is a promising method for the quantitative analysis of high-dimensional medical imaging data, which are not limited to MRI and include other imaging modalities, such as ultrasound, computed tomography (CT), and nuclear medicine imaging (e.g., positron emission tomography, PET). The radiomics process typically requires several pre-processing steps. Once the tumor is segmented, radiomics features are extracted using pre-determined mathematical methods or automatic learning techniques from the input images. The most informative features are selected, and a machine-learning model is created and tested using different classifiers. Finally, the developed model is evaluated for further analysis. Figure 1 presents a straightforward workflow that starts with the acquisition of images for MRI imaging and concludes with an evaluation after passing through segmentation, feature extraction, and selection. In neuro-oncology, radiomics has been utilized to differentiate between various conditions, recognize molecular subtypes, evaluate survival rates, and assess responses to antiangiogenic treatments [12,13]. Despite its potential to aid the diagnosis and prediction of brain tumors, there are still challenges to overcome. One of these challenges is the standardization of imaging protocols and equipment, as variations in these can affect the extracted radiomic features [14].

Additionally, overlapping radiomic features between different tumor types can make distinguishing between tumors difficult [15]. Moreover, radiomic features are typically extracted from a single time point in a patient’s imaging, which may not capture the temporal heterogeneity of tumor growth and response to treatment. Furthermore, clinical annotations on radiomics datasets can be necessary to validate the predictive power of radiomic features. In such cases, assessing the correlation between radiomic features and clinical outcomes is complex, making it challenging to validate predictive models based on these features. For instance, to develop a predictive model for treatment response using radiomic features, clinical annotations, such as patient-response data, are needed to accurately evaluate the model’s performance. These annotations are necessary to assess the accuracy of the model’s predictions, limiting its clinical utility.

Nevertheless, there has been an increase in the use of radiomics for brain tumor studies, particularly in combination with machine-learning methods. Machine learning has proven to be effective in the identification and extraction of essential characteristics of diseases, leading to a wide range of clinical uses [16]. In neuro-oncology, machine learning has yielded promising outcomes, with encouraging findings and novel opportunities for the improved care of patients affected by brain tumors [17], such as automated detection [18], differential diagnosis, grading [19] and mutation status [20], and the evaluation of the aggressiveness of tumors, as well as the prediction of treatment response, recurrence [21], and survival [22]. The application of machine learning to radiomics provides automatic, objective, and quantitative data with high efficiency, which is an improvement over the traditional radiology practice of manual annotation, which relies on trained physicians to deal with large amounts of information. In this context, this systematic review was conducted to investigate the role of radiomics and machine learning in brain tumors and their habitats for patient diagnosis and prognosis.

## 2. Materials and Methods

### 2.1. Search Strategy and Selection Criteria

A literature review was performed according to the Preferred Reporting Items for Systematic Reviews and Meta-Analyses (PRISMA) guidelines (Figure 2). The PubMed, Scopus, and WoS databases were searched to identify all potentially relevant studies from 1 January 2012 to 1 January 2023. The search query used medical subject headings (MeSH) related to AI and brain. The following search query was used on all three databases, restricted to original articles published between 2012 and 2023: 


**(RADIOMICS AND BRAIN TUMOR AND MACHINE LEARNING) OR (RADIOMICS AND BRAIN TUMOR HABITAT OR PERITUMORAL) OR (RADIOMICS AND PERITUMORAL AND MACHINE LEARNING AND BRAIN TUMOR).**


This study aimed to evaluate machine learning (ML) models for radiomics analysis in brain tumor research. A systematic search was conducted using three databases, resulting in 222 initial results. After removing duplicates, the remaining 154 articles were screened by abstract review, excluding 70. These exclusions were based on the following reasons: thirty-five were reviews, twenty were non-radiomics studies, one was a book chapter, four were histopathology studies, and ten were non-brain studies. Twelve additional articles were excluded based on specific conditions, biomarkers, or radiogenomic perspectives. The final total of eligible papers was 72; these papers were included in this review.

### 2.2. Planning and Performance of the Review

The selected articles were studied, and information relevant to the review was collected in a prearranged datasheet. The documented information included details about the study, the type of study conducted, the imaging methods used, the type of tumor analyzed, the application of the study, the dataset used, the number and type of features extracted, the method used for feature selection, the method used for classification/grading, the highest accuracy achieved, the population studied, the specifics of the training and validation, the performance of the study, the software used for radiomic- feature extraction, and the main findings of the study. The studies were then analyzed and categorized based on their purpose and the type of tumor studied.

### 2.3. Studies Corresponding Publication Year

Figure 3a provides an overview of the yearly distribution of published articles, ranging from 1 January 2012 to 31 December 2022, revealing the growing popularity of the field. No articles containing the targeted keywords were published between 2012 and 2017. However, some related content was discussed by Akbari et al. in [23], and preliminary findings were reported in [10]. 

The ground-breaking research in [10] demonstrated that machine-learning techniques could effectively predict tumor infiltration and early recurrence in glioblastoma patients using preoperative magnetic resonance images, thereby guiding targeted treatment. Moreover, the study highlighted the potential of machine-learning and pattern-analysis approaches for uncovering visually imperceptible imaging patterns to estimate the extent of the infiltration and location of future tumor recurrence.

The field has witnessed a substantial increase in published articles since 2019, with 11 and 17 publications in 2019 and 2020, respectively. The year 2021 featured the highest number of publications, with 23 articles. In 2022, 16 articles were published, indicating that research in this field has gathered momentum in recent years, experiencing a significant rise in publications from 2018 onwards. This positive trend is anticipated to persist in the coming years.

### 2.4. Characteristics of Studies

The analysis presented in this review involved the examination of 72 articles, and their characteristics are documented in Table 1. These publications were classified according to their intended purpose, which included diagnosis, prognosis, and overall survival. According to the findings presented in Figure 3b–d, most of the research (62%) was focused on gliomas, which included high- and low-grade gliomas. A substantial proportion of the studies (23%) were dedicated to predicting genetic mutations. A smaller portion of the research (14%) focused on non-glial tumors, such as brain metastases (9%) and meningiomas (5%), while only 1% of the studies were categorized as “others”.

Additionally, the analysis demonstrated that a large proportion of the studies (74%) were concerned with investigating tumors, whereas only a minority (26%) focused on exploring their habitats. Among the studies that focused on tumors, the majority (67%) aimed to diagnose the disease, while the rest (33%) investigated the prognosis. A few studies focused on differentiating the peritumoral area, with roughly equal representation from classification and survival-prediction investigations. These findings suggest that most brain tumor research has focused on gliomas, genetic mutation prediction, and tumors, with a primary emphasis on diagnosis. To improve patient outcomes and develop more effective treatments, it is crucial to persistently explore various aspects of brain tumors, including the peritumoral region and its surrounding environment, while continuing research in established areas. In addition, we calculated the RQS (research-quality score) [24] for each study, a metric used to assess the quality and rigor of research studies. It is typically calculated by evaluating various aspects of a study, such as its design, methodology, data analysis, and reporting. 

In this context, each study’s RQS score was calculated by assessing its performance in 16 sections. Each section was evaluated based on specific criteria related to study design, data collection, analysis, and reporting. The maximum achievable score was 36, indicating a higher level of research quality. This scoring system helps researchers and readers gauge individual studies’ overall quality and credibility.

**Table 1 cancers-15-03845-t001:** Characteristics of included studies. Abbreviations: NP = number of patients; NF = number of features; FS = feature selection; CM = classification method; VM = validation method, CV = cross-validation, RQS = radiomics quality score. See Abbreviation section for additional abbreviations.

Reference	Application Field	Diseases	NP (Type)	MRI Sequence	Region for Feature Extraction	Software Used	NF	FS	CM	VM	Performance	RQS
Prasanna et al., 2017 [10]	Prognosis	GBM	65(36 long-term survival, 29 short-term survival)	T2W, Gd-T1W, FLAIR	ET, NCR, PTR	MATLAB	134	12 (mRMR)	RF	3-Fold CV	CI = 0.68~0.78	56%
Shofty et al., 2018 [25]	Diagnosis	LGG	47 (26 oligodendroglia, 21 astrocytomas)	T1WGd, T2W, FLAIR	Pre-defined Lesion area of tumor	MATLAB	152	PCA	SVM, KNN, Ensemble classifier	5-Fold CV	AUC = 0.87	69%
Akbari et al., 2018 [26]	Diagnosis and prognosis	GBM	129 (74 male, 55 female)	T1WGd, T1W, T2W, T2-FLAIR, DTI	ET, non-ET, ED	CaPTk	436	Yes	SVM	10-Fold CV	AUC = 0.92	75%
Cho et al., 2018 [27]	Prognosis and survival	Glioma	285 (210 HGG, 75 LGG)	T1W, T2W, T1ce, FLAIR	ET, non-ET, ED	PyRadiomics, MATLAB	468 (3 Types)	yes (Top 5)	SVM, RF	5-Fold CV	AUC = 0.903	75%
Rathore et al. 2018 [28]	Prognosis	GBM	31	T1W, T2W, T1ce, FLAIR, DTI	ED, ET, NET	CaPTk	n/a	n/a	SVM	LOO CV	AUC =0.91	72%
Binder et al., 2018 [29]	Survival	GBM	260	T1W, T2W, T1ce, FLAIR	ET, non-ET, ED	CaPTk	1650	yes (*p* > 0.05)	Multivariate classification framework	5-Fold CV	-------	78%
Abidin et al., 2019 [30]	Diagnosis	METs and Glioma	52	T1ce, T2 FLAIR	Tumor	Amira	630	no	AdaBoost	10-Fold CV	AUC = 0.84	64%
Talamonti et al., 2019 [31]	Survival	Medulloblastoma	70	T1W TSE MDC, T2W TSE, T2W FLAIR	Necrosis, solid tumor, and oedema	PyRadiomics	----	yes	SVM	LOO-CV	------	72%
Hajianfar et al., 2019 [32]	Diagnosis	GBM	82	T1W, T2W, T1ce, FLAIR	NCR, WT, ET, ED	R, Python	7000	Top 20	Ada-Boost, DT	10-Fold CV	AUC = 0.74	81%
Hamerla et al., 2019 [33]	Diagnosis	Meningioma	147	T1W, T2W, T1ce, FLAIR	Peritumoral ED	PyRadiomics	12,733	16	SVM, RF, NLP, XGBoost	10-Fold CV	AUC = 0.97	81%
Kniep et al., 2019 [34]	Diagnosis	METs	189	T1ce, T1W, FLAIR	Multiple metastases	Python	1423	59	RF	5-Fold CV	AUC = 0.90	72%
Jeong et al., 2019 [35]	Diagnosis	GBM	25 (13 HGG, 12 LGG)	T2W FLAIR, T1W	Solid tumor	MATLAB	1689	7 (types) of delta- and radiomic features	RF	LOO CV	AUC = 0.938	69%
Wei et al., 2019 [36]	Diagnosis	GBM	105	T1ce, T2-FLAIR & ADC	Tumor & PED	R	3051	100	LR	No CV	AUC = 0.926	78%
Wening et al., 2019 [37]	Survival	GBM	211	Multimodal	ED, ET, NEC	PyRadiomics	9871	95	LR	No CV	ACC = 0.56 (long, mid, and short-term survival)	67%
Kim et al., 2019 [38]	Prognosis	GBM	83	T1W, T2W, T1ce, FLAIR, DTI, DSC	NER	ANTsR	6472	Top 10 (LASSO)	GLM	10-Fold CV	CI = 0.87	72%
Prasanna et al., 2019 [39]	Survival	Glioma	241	T1c, T2w, and FLAIR	ET, WT, TC	MATLAB	234	yes (Top 2)	CNN, RF	3-Fold CV	ST- 0.57 MT-0.63 LT-0.43	56%
Qian et al., 2019 [40]	Diagnosis	GBM & METs	412 (GBM 242, 170 METs)	T1W, T1c, T2w	Tumor & peritumoral region	PyRadiomics	1303	12	SVM, LASSO, MLP, ADaBoost	5-Fold CV	AUC = 0.95	86%
Carré et al., 2019 [41]	Diagnosis	GBM	243 (108 grade II and III gliomas, 135 grade IV GBM)	T1w-gd and T2w-flair	OED, NCR, ET	PyRadiomics	1462	91 (18 first-order and 73 second- order)	RF, NB, LR, SVM, NN	5-Fold CV	ACC = 0.82 (95% CI 0.80–0.8 5, *p* = 0.005)	72%
Shofty et al., 2020 [42]	Diagnosis	METs	53	Multi-modal	Brain lesions	MATLAB	195	50 (PCA)	SVM	5-Fold CV	AUC = 0.78	75%
Sudre et al., 2020 [43]	Diagnosis	Glioma	333 (101 LGG, 232 HGG	T2 W, FLAIR	Tumor	NiftyReg	Several	29 (Shape, histogram, Haralick)	RF	2-Fold CV	AUC = 0.80	67%
Crisi et al., 2020 [44]	Prognosis	GBM	59	T1-GRE, T2- GRE, T2FLAIR	ET, NEC	LIFEx	92	14	NB, DT, MLP	10-Fold CV	AUC = 0.84	47%
Wei et al. 2020, [45]	Diagnosis	IHPC, meningioma	292 (IHPC = 155 meningiomas = 137)	T1WI, CE- T1WI, and T2WI	TC and PED	PyRadiomics	473	64	Recursive feature elimination, RF	3-Fold CV	AUC = 0.913 (Tr), 0.914 (val)	86%
Beig et al., 2020 [46]	Survival	GBM	203	Gd-T1W, T2W, FLAIR	NCR, PED, ET	MATLAB	936	25	Cox regression	5-Fold CV	-----	81%
Lohmann et al., 2020 [47]	Early progression	GBM	34	PET	Tumor	PyRadiomics	944	4 (shape, Histogram, GLSZM)	RF	5-Fold CV	AUC = 0.79	58%
Correa et al., 2020 [48]	Diagnosis	METs	37	post-Gd T1w, T2w, and FLAIR	Lesion and lesion habitat	--------	4740 (Haralick, Gabor, Laws, CoLlAGe)	top 3 (Laws)	RF	3-Fold CV	AUC = 0.97	67%
Kumar et al., 2020 [49]	Prognosis	Glioma	285 (210 HGG, 75 LGG)	T1, T1c, T2 FLAIR	NET, NCR, ED, ET	PyRadiomics	1158	580	RF	5-fold CV	AUC = 0.97	58%
Verma et al., 2020 [50]	Survival	GBM	156	Gd-T1W, T2W, FLAIR	ET, NET, NCR	R studio	3024 (Haralick, Laws, CoLlAGe)	------	LASSO	10-fold CV	CI = 0.80	56%
Choi et al., 2020 [51]	Survival	GBM	144	T1W, T2W, T1ce, FLAIR	PED	PyRadiomics	478	7	Cox-Lasso	10-fold CV	---	75%
Yousaf et al., 2020 [52]	Survival	GBM	335 (259 HGG, 76 LGG)	T1W, T2W, T1ce and FLAIR	Tumor	MATLAB	30,632	14	RF	10-fold CV	----	53%
Zhang et al., 2020 [53]	Diagnosis and prognosis	GBM	104	T1C, T1, T2, FLAIR	ET, NCR, ED	MATLAB	180	------	SVM	No CV	ACC = 87.88%	56%
Choi et al., 2020 [54]	Diagnosis	GBM	136	T2W	Tumor & PED	PyRadiomics	107	9	Random Forest	No CV	AUC = 0.758	83%
Sakai et al., 2020 [55]	Diagnosis	Glioma	100 (22 IDH1 mutant, 78 wildtypes	FLAIR, DWI	Tumor	Olea sphere	92	----	XGBoost	5-fold CV	AUC = 0.97	67%
Demire et al. 2021 [56]	Diagnosis	GBM & METs	60 (35 GBM, 25 METs)	T1WI, T2WI, FLAIR, postcontrast T1WI	NEC, NET, ET, Oedema	Third- party	856	-----	SVM, RF, NB	5-Fold CV	AUC = 0.97	50%
Tixier et al., 2021 [57]	Survival	GBM	234	T1W	Gd -ET, NEC, NET, TC	Python	88	57	Lasso	5-Fold CV	AUC = 0.75	61%
Russo et al., 2021 [58]	Diagnosis	Glioma	56	PET	Tumor	LIFEx	44	-----	NN, RF, SVM	5-Fold CV	AUC = 0.78	50%
Yan et al., 2021 [59]	Diagnosis	GBM	41	T1ce, T1W, T2W, FLAIR	Tumor	CaPTk	841	153	RF	No CV	ACC = 81%	64%
Ye et al., 2021 [60]	Diagnosis and prognosis	GBM	285 (210 HGG, 75 LGG)	T1W, T2W, T2 FLAIR	GD-ET, PED	PyRadiomics	94	Top 30	RF, KNN, SVM, MLP, CNN	No CV	AUC = 0.65 (short-, mid-, and long-term survival)	67%
Joo et al., 2021 [61]	Diagnosis	Meningioma	454	T2W, T1ce	Tumor & PED	MATLAB	3222	Top 6	RF	10-Fold CV	AUC = 0.76	56%
Pasquini et al., 2021 [62]	Diagnosis	High-grade glioma	156	T1W, T2W, FLAIR, PWI, DWI	WT, CET, NEC, NET	MATLAB	1871	Top 15	RF	10-Fold CV	AUC = 74.2%	56%
Cao et al., 2021 [63]	Prognosis	Lower-grade glioma	102 (60 men, 42 women)	T1W, T2W, FLAIR, DWI	WT, NEC	MATLAB	56	Top 10	RF	No CV	AUC = 0.879	53%
Patel et al., 2021 [64]	Prognosis	GBM	76	CE-T1W, T2W, DWI	Whole Brain	PyRadiomics	307	6	RF, NB	10-Fold CV	AUC = 0.8	70%
Soltani et al., 2021 [65]	Diagnosis and prognosis	GBM	211	T1, T1CE, T2, and T2- FLAIR	ED, ET, NEC	PyRadiomics	3910	67	ANN, KNN, RF	No CV	ACC = 0.57 (short-, mid-, and long-term survival)	56%
Wagner et at., 2021 [66]	Prognosis	LGG	115	T2-FLAIR, Gd-T1W	Segmented tumor	PyRadiomics	851	10	RF	4-fold CV	AUC = 0.75	58%
Le et al., 2021 [67]	Diagnosis and prognosis	Glioma	120	T2-FLAIR, Gd-T1W	ET, NET, ED	CaPTk	704	13	XGBoost	LOO-CV	AUC = 0.85	61%
Kumar et al., 2021, [68]	Diagnosis	Glioma	369 (293 HGG, 76 LGG)	T2 FLAIR, T1W, postcontrast T1W and	NET, NCR, ED, ET	Python	428	----	LR, SVM, KNN, ERT	5-fold CV	AUC = 0.95	67%
Cepeda et al., 2021 [69]	Survival	GBM	203	T1CE, T1, T2, FLAIR	Tumor, peritumoral	MATLAB	15,720	----	Naive Bayes	No CV	AUC = 0.769	61%
Maliket al., 2021 [70]	Diagnosis (clinical study)	LGG & GBM	78 (42 GBM, 36 LGG)	T1ce, T2-FLAIR, DWI	PED, TC	PyRadiomics	3822	9 (RFE)	SVM, KNN, LDA, AdaBoost	LOO CV	AUC = 0.96	67%
Samani et al., 2021 [71]	Diagnosis	GBM & METs	106 (66 GBM, 40 METs)	DTI	PTR	PyRadiomics	All first-order features	Top 2% (PCA)	SVM, CNN	5-fold CV	ACC = 85%	61%
Xiao et al., 2021 [72]	Diagnosis	GBM & Brain Abscess	118 (86 GBM, 32 brain abscess)	T1W, T2W, T1ce, FLAIR	NCR, PED, TC	PyRadiomics	1004	43 (PCA)	RF, LR	5-fold CV	AUC = 0.89	56%
Gutta et al., 2021 [73]	Diagnosis	Glioma	237	T1CE, T1W, T2W, T2- FLAIR	ET, NET & ED	PyRadiomics	1284	45	SVM, RF	No CV	ACC = 87%	67%
Zhang et al., 2021 [74]	Diagnosis	Glioma	162	Gd-T1W, T1W, T2W, T2-FLAIR	TC, ED	PyRadiomics	1102	Top 10	autoML	4-fold CV	AUC = 0.951	58%
Xu et al., 2021 [75]	Prognosis	GBM	236	T1, T1-Gd, T2W, T2- FLAIR	ET, ED, NET, NCR	PyRadiomics	1320	45	Cox regression	5-fold CV	C-index = 0.64	61%
Meißner et al., 2022 [76]	Survival	METs	59	T1CE, T2W	Tumor	PyRadiomics	1316	100	SVM	10-fold CV	AUC = 0.92	67%
Shaheen et al., 2022 [77]	Survival	Glioma	178	T1W, T2W, T1ce, FLAIR	PTE, NEC, ENC	PyRadiomics	89	50	SVM	---	AUC = 0.73	61%
Deng et al., 2022 [78]	Survival	Glioma	84	T2W, T1ce, FLAIR	Tumor, NCR, ED	PyRadiomics	1316	12	RF	----	AUC = 0.879	61%
Liu et al., 2022 [79]	Prognosis	GBM	200	T1CE, T2	Tumor and peritumoral region	PyRadiomics	8412	Top 20	RF, SVM	10-fold CV	AUC = 0.91	61%
Do et al., 2022 [80]	Prognosis	GBM	53	T1W, T1Gd, T2, T2-FLAIR	NCR, PED, ET	Python	704	22	RF, SVM, XGBoost	5-fold CV	AUC = 0.93	50%
Chiu et al., 2022 [81]	Diagnosis	GBM	54	T1Gd, T2W, T2-FLAIR, T1CE	NCR, ET, PED	Python	1316	----	RF	No CV	AUC = 0.96	53%
Chen et al., 2022 [82]	Diagnosis	Meningioma	819	T1W, T2W, T1CE	Solid tumor, NCR	Python	2942	top 9	RF	No CV	AUC = 0.95	56%
Xu et al., 2022 [83]	Prognosis	Glioma	74	T1W, T2W- FLAIR, T1CE	Solid tumor	PyRadiomics	112	7	Stack, KNN, LR, RF, SVM, NB	5-fold CV	AUC = 0.76	67%
Kumar et al., 2022 [84]	Diagnosis	Glioma	285 (210 HGG, 75 LGG)	T2W, T1ce, FLAIR	NCR, ET, NET, PED	PyRadiomics	321	42	RF, DT, SVM, LR	5-fold CV	AUC = 0.975	86%
Verma et al., 2022 [85]	Survival	GBM	150	Gadolinium—T1w, T2w, FLAIR	ET, NCR	MATLAB	3792	316	----	5-fold CV	AUC = 0.78	75%
Wang et al., 2022 [86]	Diagnosis	METs	228	T1ce	Solid tumor and NCR	Python	960	548 (LASSO)	SVM	5-fold CV	AUC = 0.928	53%
Yang et al., 2022 [87]	Diagnosis	GBM	187	T1W, T2W, T1ce, FLAIR	Tumor and PED	PyRadiomics	190	Yes (LASSO)	Cox regression	10-fold CV	CI = 0.658	69%
Liu et al., 2022 [88]	Diagnosis	GBM, MET, and lymphoma	324 (134 GBM 82 Lymphoma 108 MET)	T2W, T1ce	WT, PED	PyRadiomics	8412	Top 20 (LASSO)	RF, linear, AdaBoost	10-fold CV	AUC = 0.91	62%

### 2.5. Quality Assessment

Based on the QUADAS-2 tool, this study’s assessment summary is shown in Figure 4. The risk of bias in patient selection was low in sixty-five (91%) studies, high in four (5%) studies, and unclear in three studies (4%). The risk of bias for the index test was high in 12 studies (16%) and low in 60 studies (84%). The risk of bias for the reference-standard test was low in sixty-six studies (92%), high in two studies (3%), and unclear in four studies (5%). Process and timing made the risk of bias low in 21 studies (29%), high in 33 studies (46%), and unclear in 18 studies (25%). Figure 4 shows an individual evaluation of the risk of bias and applicability. Overall, number of suitability issues was low.

## 3. Results

### 3.1. Radiomics for Glioma Grading and Differential Diagnosis

Forty-six studies are identified in this section, and all used radiomics-based machine-learning methods to classify glioma grades and types, distinguish glioma from other brain tumors or tumor mimics, or characterize tumor-progression phenotypes. For instance, Jeong et al. [35] and Prateek et al. [39] aimed to classify or differentiate high- and low-grade gliomas; and Cho et al. [27], Zenghui et al. [40], Abidin et al. [30], and Demirel et al. [56] focused on differentiating GBM from other types of brain tumor. These studies concluded that radiomics-based AI models obtained by automatic segmentation could accurately classify GBM types and distinguish them from other types with conventional sequences, reducing device- and person-dependency. While the studies had similar overall goals, there were differences in the types of imaging data used, the specific machine-learning algorithms employed, and the performance metrics and validation methods used to evaluate the effectiveness of the approaches—for example, Prateek et al. [39] utilized a convolutional neural-network framework to integrate radiomic texture features, while Jeong et al. [35] used delta-radiomic features from dynamic-susceptibility contrast-enhanced MRI. SVM [89], random forest (RF) [90], artificial neural networks (ANN), logistic regression (LR), and eXtreme gradient boosting (XGBoost) are commonly used for classification. Again, in [59,75], radiomics approaches were used on preoperative multimodal MRI data to characterize tumor progression phenotypes. The variations among these studies lie in their areas of focus, sample sizes, and machine-learning-classifier methods. For glioma grading, [43,49,56,61,67,68,91] most of the studied papers applied K-fold (K from 2 to 10) cross-validation (CV). The performance of these models was measured through the area under the curve (AUC) values, indicating varying levels of efficacy. Furthermore, distinct evaluation metrics, such as accuracy, sensitivity, and specificity, were also employed in the studies.

### 3.2. Radiomics for Non-Glial Tumors 

Three studies on meningioma diagnosis are reviewed in this section. The first study [33], aimed to differentiate between Grade I, II, and III meningiomas using radiomics features extracted from multiparametric MRI. The study included one hundred and thirty-eight patients from five international centers, and four machine-learning classifiers were used to score the selected features. The second study [45], developed an integrated diagnostic tool called the IHPC and the Meningioma Diagnostic Tool (HMDT) to distinguish intracranial hemangiopericytoma from meningioma using a multihabitat-based radiomics strategy. The study included 292 patients with complete clinical–radiological and histopathological data. The HMDT displayed remarkable diagnostic ability, with AUC values of 0.985 and 0.917 in the training and validation cohorts, respectively. The third study [82] developed an automated segmentation approach for meningioma utilizing deep learning. Subsequently, it evaluated the ability of the approach to differentiate between different types of meningioma before surgery using radiomic features. This multicenter study retrospectively examined MR images from 609 patients. The meningioma segmentation was conducted using a modified-attention U-Net, and L1-regularized logistic regression models was created separately to distinguish between Grade I and Grade II/III meningiomas using manual and automated segmentations.

All these studies involved the analyses of MRI images and included a large number of patients from multiple centers. The first two studies focused on differentiating distinct subtypes of meningiomas, while the third focused on segmentation and differentiation between meningiomas of different grades. These studies demonstrate the potential of radiomics and machine-learning techniques to improve the accuracy of meningioma diagnosis, which could have significant implications for patient outcomes and treatment planning. 

### 3.3. Radiomics for Survival Prediction 

Radiomics methods based on ML are new developments in the prediction of patient survival, with no prior exploration of this topic before 2019. Weninger et al. [37] began exploring various radiomics-based methods for predicting survival based on the BraTS dataset [92]. Their study evaluated different radiomic approaches to predicting brain tumor patients’ survival, including a linear-regression baseline based on age only. The use of radiomics showed promise for patients with subtotal resection. However, it performed poorly for patients with gross total resection, which explains the poor overall performance of radiomics-based approaches in the BraTS dataset. Two studies [40,77] focused on the progression-free survival of patients with glioblastoma.

In contrast, other studies concentrated on classifying overall survival or calculating survival time using the radiomics model [68] or radiomics score [36]. Deng et al. [68] applied a multiregional model that extracts radiomic features from multiple ROIs within tumors and evaluated the effectiveness of four multiregional radiomic models in predicting overall survival. The random survival forest (RSF) algorithm was used to perform the survival analyses. The naive Bayes classifier achieved the best results in the test data set, with an AUC of 0.769 and a classification accuracy of 80%.

### 3.4. Radiomics for Brain-Habitat Analysis

In recent years, there has been growing interest in the use of radiomics features to analyze tumor habitats, i.e., peritumoral environments. In this regard, authors employ different terms to describe the surrounding area of a tumor, including the lesion habitat, brain habitat, multi-habitat, and subcompartments or microenvironments, which can form peritumoral edema or be affected by tumor growth. 

The peritumoral region, also known as the tumor microenvironment, plays a crucial role in glioblastoma’s recurrence [28,88] and provides valuable insights for diagnosis and prognosis. This region is characterized by various molecular and cellular changes contributing to tumor growth, invasion, and treatment resistance. Exploring the peritumoral region can reveal important clues about the underlying mechanisms driving GBM recurrence.

One of the key features of the peritumoral region is its heterogeneity. In diffuse gliomas and glioblastoma, it consists of brain regions infiltrated by distinct subpopulations of tumor cells with varying genetic and epigenetic alterations. This heterogeneity leads to differences in cellular behavior, including invasive potential and resistance to therapy [36]. Researchers can identify specific biomarkers associated with recurrence and aggressiveness by analyzing the molecular and genetic [35] profiles of cells in the peritumoral region. This information can aid in developing targeted therapies and personalized treatment strategies.

Despite the varying terminology across studies, the underlying concept is the same: the analysis of the characteristics of the tumor’s surrounding tissues and the tumor itself can provide valuable information for diagnosis, treatment planning, and the prediction of patient outcomes. For instance, Verma et al. [50] analyzed lesion heterogeneity on clinical MRI to stratify patients into low- and high-risk categories based on progression-free survival, while Neha Beig et al. [46] developed a survival-risk score for predicting progression-free survival in glioblastoma using radiomic features extracted from routine MRI scans. Other studies investigated the potential of radiomics as imaging biomarkers for glioblastoma patients [51], developed a radiomic-pipeline to differentiate radiation necrosis from tumor recurrence in brain-metastases patients [28,48], and utilized multi-habitat radiomics features and clinical–radiological information to distinguish between different tumor types [45]. In [71], deep learning was employed to analyze the microstructure of the surrounding tissue to identify distinct tumor characteristics. The study’s outcomes revealed that traditional machine-learning techniques cannot find subtle features that a convolutional neural network can detect. This approach may hold potential for accurately distinguishing between various types of brain tumors. Other studies [50,75,79,81,85,87] also analyzed this category and focused on glioblastoma tumors. Some of them [79,81] investigated the classification or grading of glioblastoma tumors, aiming to understand better the molecular and genetic characteristics that may contribute to their development and progression. This could help to identify different glioblastoma-tumor subtypes and develop tailored treatment approaches. Overall, radiomics research is used to investigate the significance of habitat features in anticipating patient outcomes and treatment response. Furthermore, the relationship between radiogenomic associations and molecular-signaling pathways is under study to better understand these features’ biological basis.

### 3.5. Radiomics for Genetic-Mutation-Status Prediction

Our analyses identified a considerable amount of research highlighting the potential of radiomics methodologies for predicting important genetic biomarkers in gliomas, such as the epidermal growth factor receptor (EGFR), isocitrate dehydrogenase 1 (IDH1), O^6-methylguanine-DNA methyltransferase (MGMT), and v-Raf murine sarcoma viral oncogene homolog B (BRAF) mutations, using various MRI contrasts, radiomic features, and machine-learning algorithms (in the so-called radiogenomic field). These techniques offer significant opportunities for improving patient outcomes. This is demonstrated by three articles published in 2018, the first two of which [26,29] examined the impact of EGFR mutations on the overall survival rates and the efficacy of EGFR-targeted therapies. Another study [25], aimed to assess the effectiveness of radiomics in categorizing patients with low-grade gliomas based on their IDH1 mutations and 1p/19q codeletion status. 

The analyses also highlight four significant advances [54,55,63,93] that focus on the use of radiomic features and machine-learning methods to predict IDH1-mutation status in gliomas, and five MGMT-focused articles [32,36,44,62,80] that utilize radiomics to predict the methylation status of the MGMT gene promoter in glioblastoma multiforme patients. Two articles used radiomics approaches and machine-learning techniques to predict the molecular markers (IDH1 and MGMT) and prognoses of glioblastomas and gliomas [74,94]. The first study [94] developed a prognostic model using clinical and radiomics features for GBM patients, with promising results in the prediction of molecular markers, such as IDH mutation, MGMT methylation, and EGFR amplification. The second study [74], aimed to determine the ability of glioma radiomics features on MRI to predict overall survival and progression-free survival. 

Wagner et al. [66] and Meißner et al. [76] demonstrated the feasibility of using radiomics features extracted from MR images to predict the BRAF status of pediatric low-grade gliomas and intracranial BRAF-V600E-mutation status in patients with melanoma brain metastases, respectively. Additionally, the analyses cite a study [95] that developed an automated machine learning (autoML) approach using radiomics features to predict H3-K27M-mutation status in midline gliomas in the brain. 

These studies suggest that the radiomics-based prediction of genetic mutation status holds promising clinical relevance for aiding clinical decision-making in managing patients with brain tumors. Additionally, radiomics approaches can aid in preoperative stratification for targeted therapies and improve patient outcomes. 

### 3.6. Frequently Selected Radiomics Features

Radiomics features have the potential to provide valuable information about the tissue properties within a region of interest (ROI). Depending on their information, these features can be categorized according to size and shape, texture, intensity, and wavelets. Among the various radiomics features, some of the most frequently used in recent studies are the gray-level co-occurrence matrix (GLCM) [41], the gray-level run-length matrix (GLRLM), the gray-level size-zone matrix (GLSZM), Haralick features [43], Laws features, Gabor features, and histogram-based features.

In addition to the shape- and intensity-based features, several other radiomic features, including Haralick, Gabor, Laws, and the co-occurrence of local anisotropic gradient orientations (CoLlAGe) [46,48,50], are commonly used in medical imaging analysis. Haralick features describe the spatial relationship between pixel values based on gray-level co-occurrence matrices. Gabor features use filters with different frequencies and orientations to capture image texture information. Laws features use predefined filter masks to capture different texture patterns at various scales. The CoLlAGe [39] features consider both the spatial arrangement and the intensity levels of pixel values in an image, providing a robust measure of texture heterogeneity. These radiomic shape- and intensity-based features can provide valuable insights into tumor heterogeneity and may have potential applications in diagnostic and prognostic settings.

The analysis of these features can provide insights into a tissue’s heterogeneity, complexity, and spatial distribution within a ROI, which is crucial in the diagnosis, prognosis, and treatment planning of various diseases. However, the selection of radiomics features may vary depending on several factors, including the type and location of the cancer studied and the imaging modality employed. Our analysis of the 72 selected articles identified the most frequently used radiomics features, which are further described in Appendix A.

### 3.7. Evaluation Metrics

Researchers who work on ML models for brain tumor diagnosis and prognosis commonly use a range of evaluation metrics. These metrics provide an objective way of assessing the performance of these models in identifying and classifying different types of brain tumors, such as glioblastoma, metastasis, and meningioma.

One of the most basic and commonly used metrics is accuracy, which measures the proportion of correctly classified instances. In brain tumor diagnosis, accuracy is used to measure how often a model correctly identifies the type of tumor. Sensitivity and specificity are other commonly used metrics that measure the proportion of true positives and true negatives, respectively, and indicate how well a model can distinguish between different types of brain tissue. Graphical representations of evaluation metrics, such as the receiver operating characteristic (ROC) and the precision–recall curve, are also commonly used. The ROC curve shows the trade-off between sensitivity and specificity for different classification thresholds.

In contrast, the precision–recall curve visually represents the balance between precision and recall for different classification thresholds, mainly when the dataset is unbalanced. The area under the ROC curve (AUC) is another commonly used metric for binary classification problems. A higher AUC value indicates better performance in distinguishing between positive and negative cases. The F1 score is a metric that evaluates the balance between precision and recall, while a confusion matrix provides a table summarizing the overall performance of a classification model. 

Along with the other metrics described above, one commonly used metric to assess the performance of radiomics models in predicting the overall survival of brain tumor patients is the concordance index (C-index), which measures a model’s ability to distinguish between patients with different survival times. The C-index is determined by comparing the expected and actual survival probability. The Kaplan–Meier estimator, a non-parametric method for estimating survival probabilities over time, was also found in many papers. The Kaplan–Meier estimator is often used to compare the survival curves of different patient groups. Another metric used for evaluating a model’s predictive accuracy is Harrell’s C statistic. This statistic is a variant of the C-index and considers both a model’s discrimination and calibration.

These metrics can be used individually or in combination to evaluate the performance of radiomics models for overall survival prediction in brain cancer patients. However, the choice of metrics depends on the specific research question and the available data.

## 4. Discussion

### 4.1. Promises of Radiomics and Machine Learning for Brain Tumor Analysis

Radiomics-based approaches have shown great potential in predicting patient survival and differentiating between subpopulations of patients with various diseases, including brain tumors. In the case of brain tumor patients, ML-based radiomics approaches have been explored in recent years, and studies have evaluated the effectiveness of various radiomic approaches. The addition of location-based radiomic features to classifier models has improved prediction accuracy in some cases. Furthermore, radiomics-based approaches have been used to analyze brain tumor habitats to identify radiogenomics-based survival risk in glioblastoma cases. The radiomic features extracted from tumor subcompartments have been used to develop radiomic risk scores (RRS) for predicting progression-free survival (PFS) in GBM patients.

Radiomics also show promise in predicting treatment response and assessing tumor aggressiveness. By quantifying tumor heterogeneity and capturing subtle changes in the tumor microenvironment, radiomic features can serve as imaging biomarkers that are correlated with treatment response and prognosis. Multiple research studies have showcased the potential of radiomics for forecasting treatment response in brain tumors, aiding clinicians in identifying suitable treatment options and enabling personalized planning. Additionally, Lennart et al. [96] highlight the impact of glioblastoma’s molecular diversity and communication with the microenvironment on therapy resistance, underscoring the need for enhanced treatment strategies. Furthermore, the significance of glioma-associated macrophages (GAMs) in increasing tumor progression, drug resistance, and immunosuppression highlights the necessity for targeted therapies to improve patient survival, as emphasized in another study [97]. 

Another promising application of radiomics in brain tumor analysis is its ability to provide valuable insights into the tumor habitat. Tumors interact with the surrounding brain tissue, leading to changes in the peritumoral region. Radiomics can capture these changes by analyzing the spatial relationships and textural patterns within tumors and their habitat. This information can aid in identifying regions of infiltration, assessing the extent of tumor spread, and evaluating the impact on neurological functions. By combining radiomics with functional imaging techniques, such as functional MRI (fMRI), researchers can also investigate the functional connectivity and network alterations in tumor habitats, providing a more comprehensive understanding of a tumor’s impact on the brain, as recently demonstrated by Krishna et al. in their seminal paper on the correlation between glioblastoma’s remodeling of human-brain neural circuits and decreased patient survival [98].

The studies included in this review highlight the potential of radiomics for improving the accuracy and efficiency of tumor grading and diagnosis. Most studies focus on conventional MRI sequences, such as T1-weighted, T2-weighted, and contrast-enhanced T1-weighted images, which offer anatomical details and aid in tumor localization and characterization. However, they may not capture subtle microstructural changes or functional alterations associated with tumors. The integration of advanced modalities, such as diffusion-weighted imaging (DWI) and diffusion tensor imaging (DTI), provides insights into tissue microstructure, cellularity, and white-matter integrity. The application of DWI is used to assess tumor cellularity and detect restricted diffusion, indicating areas of high cellular density or necrosis. At the same time, DTI maps tumors’ infiltration into surrounding brain tissue by characterizing the water-diffusion directionality. Susceptibility-weighted imaging (SWI) helps to assess compounds that create susceptibility effects on MRI, such as blood products and calcium. The fMRI techniques, including blood-oxygen-level-dependent (BOLD) imaging and task-based paradigms, assess brain-activation patterns and connectivity, aiding preoperative planning and minimizing postoperative deficits. Perfusion imaging techniques, like dynamic susceptibility contrast (DSC) and arterial spin labeling (ASL), provide information about tumor vascularity, blood flow, and perfusion, which is relevant for grading and treatment-response evaluation. 

To summarize, the application of radiomics-based approaches has demonstrated exceptional potential in predicting patient survival, distinguishing subpopulations, and examining the characteristics of brain tumor habitats. Radiomics enables the quantification of tumor heterogeneity and the detection of subtle alterations in the tumor microenvironment, making it a valuable tool for imaging biomarkers that are correlated with treatment response and prognosis. Integrating radiomics with advanced imaging modalities, such as DWI, DTI, SWI, fMRI, and perfusion imaging, provides a comprehensive understanding of tumors’ properties, infiltration patterns, and functional changes.

### 4.2. Research Gaps and Future Challenges

Radiomics still faces several challenges and includes research gaps that must be addressed. Standardization is a crucial challenge in radiomics, as there is no agreed-upon protocol for image acquisition, pre-processing, feature extraction, and analysis. This lack of standardization can lead to variability in results, making it difficult to draw meaningful conclusions from different studies.

Another challenge is the limited sample size of the datasets used in radiomics research. Many studies use small datasets, leading to overfitting and limiting the generalizability of the results. Additionally, radiomics involves the analysis of large and complex datasets, which can be challenging to manage and analyze. To overcome this challenge, advanced computational methods and tools are required.

Reproducibility is another critical issue in radiomics research. Reproducibility refers to the ability to replicate the results of a study using different datasets or methods. There is a need for the standardization and validation of radiomics methods to improve the reproducibility of results.

Moreover, integrating radiomics data with clinical data is essential for improving cancer diagnosis, prognosis, and habitat studies. However, several challenges need to be overcome, including the need for standardized data formats and methods for data integration.

Finally, the validation of radiomics models ensures their clinical relevance and usefulness. Large-scale multicentre studies are needed to validate radiomics models and demonstrate their clinical utility.

Although radiomics has the potential to revolutionize cancer diagnosis and prognosis, as well as habitat studies, there is a need for standardization, advanced computational methods, large-scale datasets, and validation studies to realize this potential fully. Addressing these challenges will enable radiomics to provide valuable insights into cancer biology and help clinicians make more informed decisions about cancer treatment.

## 5. Conclusions

In recent years, radiomics approaches have shown great potential for brain tumor and tumor-habitat studies. Advancements in radiomics approaches have enabled the extraction of more complex and diverse features from medical images, improving the characterization of brain tumors and their microenvironments. This has led to a better understanding of tumor heterogeneity, which is critical for guiding treatment decisions and predicting patient outcomes. Radiomics-based approaches have also shown promise for non-invasive diagnosis, monitoring, and prediction of treatment response in brain tumors. 

Overall, the advancements in the radiomics approach in brain tumor and tumor habitat studies are promising to improve brain tumor diagnosis, treatment, and monitoring. However, further research is needed to fully understand the potential of radiomics-based approaches and develop robust and standardized methods for their implementation in clinical practice.

## Figures and Tables

**Figure 1 cancers-15-03845-f001:**
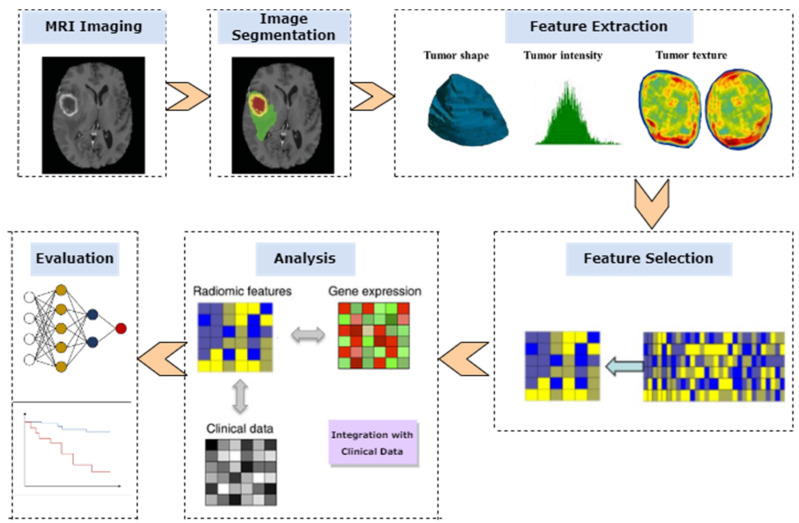
Radiomics workflow: A diagram illustrating the various steps involved in the radiomics workflow, starting with image acquisition for MRI imaging and ending with evaluation, after passing through segmentation, feature extraction, and selection.

**Figure 2 cancers-15-03845-f002:**
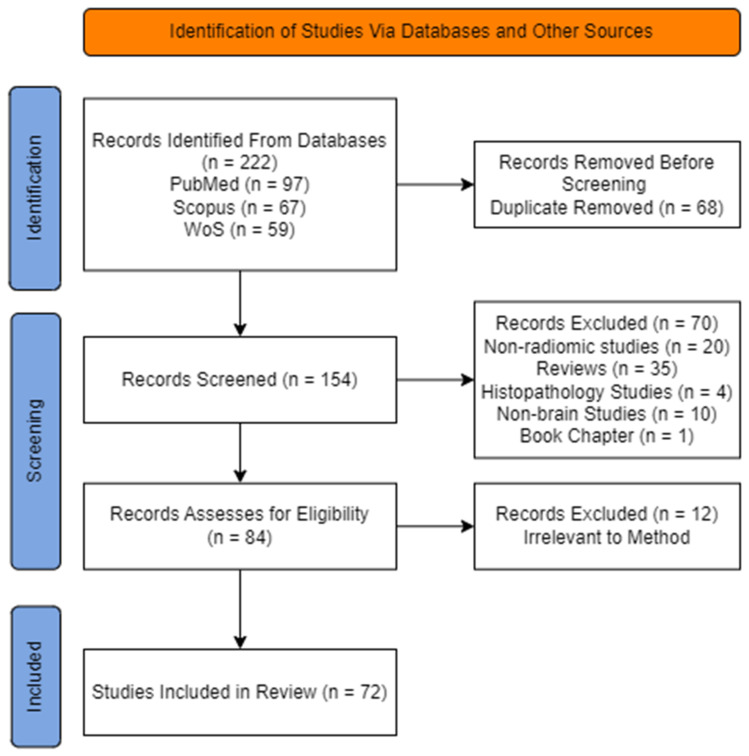
The PRISMA diagram shows the screening and selection of relevant papers.

**Figure 3 cancers-15-03845-f003:**
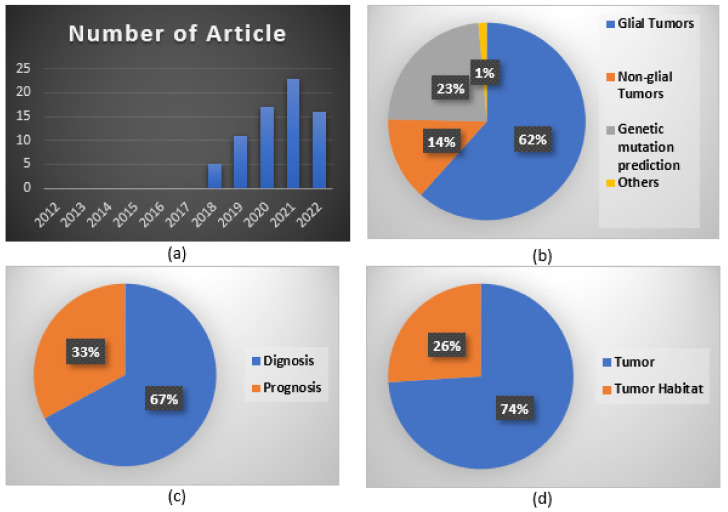
(**a**) Bar chart of the number of articles included in this review according to their publication year. Three pie charts (**b**–**d**) are presented, depicting the number of articles and focusing on types of study, application area, and subject.

**Figure 4 cancers-15-03845-f004:**
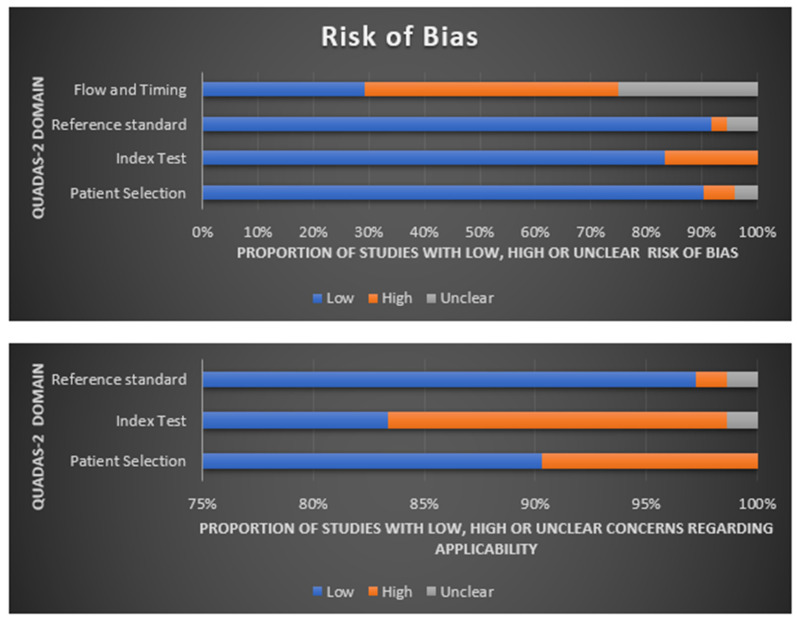
Summary of QUADAS-2 assessments of included studies.

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
