# Peer review of "Radiomics and Machine Learning in Brain Tumors and Their Habitat: A Systematic Review"

_cancers, 2023, doi:10.3390/cancers15153845_

Round 1

Reviewer 1 Report (Previous Reviewer 3)

The manuscript has been revised accordingly. Is it good now.

 Minor editing of English language required

Reviewer 2 Report (Previous Reviewer 1)

The authors have successfully responded to the suggestions made, thereby improving the overall quality of the manuscript.

This manuscript is a resubmission of an earlier submission. The following is a list of the peer review reports and author responses from that submission.

Round 1

Reviewer 1 Report

In this study, the authors have conducted a comprehensive systematic review focusing on the combination of machine learning and radiomic features in the study of brain tumors and their habitats. This is an area of immense clinical relevance, and the frequency of publications in this field is steadily increasing. The authors have employed an appropriate methodology for their systematic review, and their selection of bibliographic references demonstrates careful consideration.

To further enhance the rigor of their study, I would suggest that the authors consider incorporating the use of the QUADAS-2 tool as an instrument for evaluating the quality of diagnostic accuracy studies within the relevant publications. The QUADAS-2 (Quality Assessment of Diagnostic Accuracy Studies) tool is a widely recognized and accepted framework for assessing the risk of bias and applicability of studies in the field of diagnostic accuracy. By employing this tool, the authors can provide a more comprehensive evaluation of the quality and validity of the included studies, thereby strengthening the overall robustness of their findings.

In addition to incorporating the QUADAS-2 tool, it would be valuable for the authors to include the Radiomics Quality Score (RQS) of the different articles. The RQS is a metric proposed by Lambin et al. in their seminal publication "Radiomics: The bridge between medical imaging and personalized medicine" (2017). This score evaluates the methodological quality and reporting completeness of radiomics studies, taking into account various aspects such as study design, feature extraction, statistical analysis, and validation. By including the RQS for each article, the authors can provide a standardized assessment of the methodological rigor and reporting quality of the studies included in their systematic review.

By incorporating these additional measures, the authors can further strengthen the reliability and comprehensiveness of their systematic review. The use of the QUADAS-2 tool would enable a more rigorous evaluation of the included studies, while the inclusion of the RQS would provide insights into the methodological quality and reporting standards of the articles. Overall, these additions would enhance the value and impact of the study, allowing for a more robust assessment of the combination of machine learning and radiomic features in the study of brain tumors and their habitats.

Author Response

Thank you for your valuable suggestion. We wholeheartedly agree with your recommendation. In our study, we employed the QUADAS-2 (Quality Assessment of Diagnostic Accuracy Studies) tool to assess each paper. This tool allowed us to evaluate the risk of bias and applicability concerns in a systematic and standardized manner.

To ensure the accuracy and reliability of our assessments, we assigned an independent assessor(added as a co-author now) to evaluate each paper using the QUADAS-2 tool. We then compared the assessor's evaluation with our own assessment, and any discrepancies were resolved through thorough discussions.

Following this rigorous process, we finalized the QUADAS-2 assessments. Additionally, we completed the RQS (Research Quality Score) scoring, adhering to the specific requirements of the RQS scoring. we add the percentages of each study with an additional column in the characteristic table.

Reviewer 2 Report

This is a systematic review of the role of radiomics and machine learning in brain tumors and their habitat. It highlights the value of radiomics in the evaluation of tumor type, the genetic features of tumors, treatment planning and pronostic.

This paper is well-written and easy to follow. The introduction is adequate. The conclusion is supported by the data. The methodology is sound.

The resolution of figure 3 should be higher.

Author Response

Thank you for your feedback on our paper. We are grateful for your positive comments regarding the clarity and coherence of our manuscript.

We acknowledge and agree with your suggestion to enhance the resolution of Figure 3. In response to this, we have already replaced the figure with a higher-resolution version. This modification aims to improve the visual clarity and readability for the readers.

Reviewer 3 Report

This topic is very interesting, but some points need to be revised before the paper can be considered for publication:

- "In this context, this systematic review has been conducted to investigate the role of radiomics and machine learning in brain tumors and their habitat for patient diagnosis and prognosis." What does this review add to the previous review about the same topic?

- "studies from 1 January 2012 to 1 January 2023" Can the authors explain why they chose this time frame? is there any particular reason?

- Figure 3 shows that before 2018, no papers were published about this topic. Discuss this point.

- "By quantifying tumor heterogeneity and capturing subtle changes in the tumor microenvironment, radiomic features can serve as imaging biomarkers that correlate with treatment response and prognosis." Some previous very important and recent papers should be considered in the discussion section. In Pubmed: -- PMID: 37218976 --  doi: 10.3390/neurolint15020037 -- -- PMID: 35741816 --  DOI: 10.3390/genes13061054  -- DOI: 10.1007/s10555-021-09997-9

- "Additionally, radiomics approaches can aid in preoperative stratification for targeted therapies and improve patient outcomes." What do the authors mean by these sentences?

- "In conclusion, radiomics-based approaches have demonstrated outstanding potential in... " This is part of the discussion. Revise this part. Possibly insert this part in the conclusion.

Minor editing of English language required.

Author Response

Thank you for considering our paper and for your valuable feedback. We appreciate your interest in the topic and are glad to hear that you find it interesting. I have tried to meet your comment and listed here point by point.

- "In this context, this systematic review has been conducted to investigate the role of radiomics and machine learning in brain tumors and their habitat for patient diagnosis and prognosis." What does this review add to the previous review about the same topic?

This systematic review significantly contributes to the existing literature on the role of radiomics and machine learning in brain tumors by focusing not only on the tumors themselves but also on their surrounding environment, known as the tumor habitat. Unlike previous reviews that primarily analysed the microenvironment or peritumoral region, this review comprehensively covers all aspects of the brain, including different types of tumors and their habitats, from a machine learning-based radiomics perspective.

By dividing the analysis into various categories such as glial tumors, non-glial tumors, survival prediction, genetic mutation prediction, and brain habitat, this review provides a comprehensive understanding of the subject matter. This approach allows for a more nuanced examination of the impact of radiomics and machine learning techniques in diagnosing and predicting the prognosis of brain tumor patients.

Furthermore, this systematic review identifies potential research gaps in the field and offers suggestions for future steps in the diagnosis and prognosis of brain tumor patients. By addressing these gaps and providing recommendations, the review aims to guide further research in utilizing radiomics and machine learning in brain tumor analysis.

- "studies from 1 January 2012 to 1 January 2023" Can the authors explain why they chose this time frame? is there any particular reason?

We appreciate the reviewer's query regarding the chosen time frame for our systematic review. The decision to include studies from January 1, 2012, to January 1, 2023, was based on several considerations and methodological justifications.

Initially, our intention was to conduct a comprehensive analysis of the research conducted in the field of radiomics and machine learning in brain tumors over a period of 10 years. As the review was initiated in the middle of 2022, we aimed to cover studies published from January 2012 to January 2022, thereby encompassing a decade of relevant research.

However, as the review progressed, we encountered some notable papers that were published after the initial time frame. These recent papers were found to be highly relevant and provided valuable insights into the topic, which motivated us to extend the time frame to include the year 2023. By including studies published up until February 2023, we aimed to incorporate the most up-to-date and cutting-edge research findings in our review.

It is crucial to consider that research in rapidly evolving fields such as radiomics and machine learning can yield significant advancements within a short span of time. By including studies up to 2023, we aimed to capture the most recent developments and ensure the comprehensiveness and relevance of our systematic review.

- Figure 3 shows that before 2018, no papers were published about this topic. Discuss this point.

Our intention was to specifically identify machine learning papers that focus on the brain habitat. However, during our search, we did not come across any articles that precisely matched our specified keywords and criteria before 2018.

- "By quantifying tumor heterogeneity and capturing subtle changes in the tumor microenvironment, radiomic features can serve as imaging biomarkers that correlate with treatment response and prognosis." Some previous very important and recent papers should be considered in the discussion section. In Pubmed: -- PMID: 37218976 --  doi: 10.3390/neurolint15020037 -- -- PMID: 35741816 --  DOI: 10.3390/genes13061054 

We would like to express our gratitude to the peer reviewer for suggesting additional papers to consider for inclusion in the discussion section of our manuscript. We have thoroughly reviewed the provided references and have carefully considered their relevance and contribution to our research.

After a comprehensive evaluation, we agree that the first and third papers mentioned, PMID: 37218976 (DOI: 10.3390/neurolint15020037) and DOI: 10.1007/s10555-021-09997-9, respectively, are highly pertinent to our study and will significantly enhance the discussion section. These papers align with the scope and objectives of our review, and their findings and insights are directly applicable to the topics addressed in our research.

However, regarding the second paper mentioned, PMID: 35741816 (DOI: 10.3390/genes13061054), we respectfully disagree with its direct relevance to our study and its potential to amplify our concept. After carefully examining the paper, we found that its focus and content are not directly aligned with the main themes and objectives of our systematic review.

We believe that the inclusion of the first and third papers will sufficiently strengthen and enrich the discussion section, providing meaningful insights and supporting our research findings. We are grateful to the peer reviewer for their suggestions, and we will ensure that appropriate acknowledgments and citations are provided for the valuable contributions of these papers in the revised version of our manuscript.

Once again, we sincerely appreciate the reviewer's efforts and insightful suggestions, which have undoubtedly improved the quality and comprehensiveness of our work.

- "Additionally, radiomics approaches can aid in preoperative stratification for targeted therapies and improve patient outcomes." What do the authors mean by these sentences?

Preoperative stratification refers to the process of categorizing patients into different groups based on specific characteristics or criteria before surgery. In the case of brain tumors, preoperative stratification can help identify patients who are more likely to respond to specific targeted therapies. These therapies are designed to target specific molecular or genetic abnormalities present in the tumor, providing a more personalized and effective treatment approach.

By applying radiomics approaches to analyse medical images, relevant features and patterns can be extracted. These features may include texture, shape, intensity, or spatial characteristics of the tumor and its surrounding tissue. This information can then be used to identify specific biomarkers or signatures associated with treatment response or prognosis.

We believe that incorporating radiomics approaches into the preoperative stratification process can provide additional insights that help healthcare professionals make more informed decisions regarding the most appropriate targeted therapies for individual patients. By tailoring the treatment strategy based on radiomics findings, it is expected that patient outcomes can be improved, leading to more effective and personalized care for individuals with brain tumors.

- "In conclusion, radiomics-based approaches have demonstrated outstanding potential in... " This is part of the discussion. Revise this part. Possibly insert this part in the conclusion.

Agree and done.